# Efficient TGF-β1 Delivery to Articular Chondrocytes In Vitro Using Agro-Based Liposomes

**DOI:** 10.3390/ijms23052864

**Published:** 2022-03-05

**Authors:** Émilie Velot, Kamil Elkhoury, Cyril Kahn, Hervé Kempf, Michel Linder, Elmira Arab-Tehrany, Arnaud Bianchi

**Affiliations:** 1IMoPA (Molecular Engineering and Articular Physiopathology), CNRS (French National Centre for Scientific Research), Université de Lorraine, F-54000 Nancy, France; emilie.velot@univ-lorraine.fr (É.V.); herve.kempf@inserm.fr (H.K.); 2LIBio (Laboratoire d’Ingénierie des Biomolécules), Université de Lorraine, F-54000 Nancy, France; kamil.elkhoury@univ-lorraine.fr (K.E.); cyril.kahn@univ-lorraine.fr (C.K.); michel.linder@univ-lorraine.fr (M.L.)

**Keywords:** liposomes, chondrocytes, transforming growth factor-β1, drug delivery system, joint regenerative medicine

## Abstract

The low efficiency in transfecting rat- and human-derived chondrocytes have been hampering developments in the field of cartilage biology. Transforming growth factor (TGF)-β1 has shown positive effects on chondrocytes, but its applications remain limited due to its short half-life, low stability and poor penetration into cartilage. Naturally derived liposomes have been shown to be promising delivery nanosystems due to their similarities with biological membranes. Here, we used agro-based rapeseed liposomes, which contains a high level of mono- and poly-unsaturated fatty acids, to efficiently deliver encapsulated TGF-β1 to rat chondrocytes. Results showed that TGF-β1 encapsulated in nano-sized rapeseed liposomes were safe for chondrocytes and did not induce any alterations of their phenotype. Furthermore, the controlled release of TGF-β1 from liposomes produced an improved response in chondrocytes, even at low doses. Altogether, these outcomes demonstrate that agro-based nanoliposomes are promising drug carriers.

## 1. Introduction

Chondrocytes are very specific cells that form cartilage. When chondrocytes are present in joints, they differentiate and express in their mature functional form, named as articular phenotype, that is primarily identified by the expression of type II collagen (Col 2), aggrecan (ACAN), and Sox-9 (SRY (Sex-Determining Region Y)-Box 9) genes [1,2,3]. Changes in the extracellular matrix (ECM) composition and structure, such as supramolecular collagen network destabilization or proteoglycan content depletion, can lead to the loss of the articular phenotype [4,5].

In vitro transfection methods of chondrocytes either possess a very low transfection efficiency or high toxicity, whereas in vivo strategies are limited by the lack of blood circulation because of the avascular nature and poor self-repair capacity of articular cartilage. As for viral and nonviral vectors, limitations also exist. Viral vectors are usually more efficient, but suffer from safety issues, such as immunogenicity and oncogenesis. On the other hand, nonviral chemical-based vectors are generally regarded as more safe, easy-to-use alternatives, but possess low chondrocyte transfection efficiencies [6].

Ultimately, agro-based rapeseed liposomes were recently proposed as a new, safe and effective alternative for chondrocyte transfection [6]. Liposomes are mainly composed of amphiphilic phospholipids, and due to the similarities between their lipid bilayer and biological membranes, they are a promising delivery system, which can safely deliver and control the release of drugs and bioactive molecules [7,8,9]. Furthermore, bioactive agents are well protected once encapsulated in liposomes from the external harsh in vivo environment [10,11,12]. This could solve the short half-life, low stability and poor penetration into cartilage of transforming growth factor (TGF)-β1, which has shown positive effects on chondrocytes [13,14,15,16] by binding to their receptors and activating signaling pathways among which the best known are extracellular signal-regulated kinase (ERK), p38-mitogen-activated protein kinase (p38-MAPK) and Smad [17,18,19]. Indeed, TGF-β1 is known to retain chondrocyte articular phenotype by inducing the genes coding for Col 2, ACAN and Sox-9 [16,20], as well as the release of inorganic pyrophosphate (PPi) [21,22]. Additionally, the lipidic composition of rapeseed liposomes is of great interest, since they possess a high level of mono- and poly-unsaturated fatty acids, mainly linoleic acids (ω-6) and linolenic acids (ω-3), essential for human health and cannot be produced by the human body [23,24,25].

The aim of our work was to demonstrate that rapeseed liposomes, as vectorization systems, are able to encapsulate and deliver an active molecule that can influence cell phenotypes in vitro. In this study and for the first time, rapeseed liposomes were evaluated as a potential safe and efficient carrier of TGF-β1 to rat healthy chondrocytes. The physicochemical properties of these agro-based liposomes, in addition to the cytotoxicity, release kinetics, and biofunctionality of TGF-β1-loaded liposomes were characterized. The release of TGF-β1 from liposomes was revealed to be progressive, with the growth factor remaining active, which validates our proof of concept. Overall, these results obtained with TGF-β1 showed that agro-based nanoliposomes are very suitable delivery systems for growth factors to maintain articular chondrocyte phenotype and have a promising therapeutic potential in joint regenerative medicine.

## 2. Results

### 2.1. Rapeseed Liposomes Physicochemical Properties and Morphology

The measured rapeseed nanoliposomes average diameter, ζ-potential and polydispersity index (PDI) measured using the dynamic light-scattering (DLS) technique were found to be 135.5 ± 1.45 nm, −40.3 ± 0.5 mV and 0.19 ± 0.01, respectively. As the PDI value is <0.3, the size distribution of the sample is considered as narrow (Figure 1A).

Morphological studies using transmission electron microscopy (TEM) showed that liposomes were of an ellipsoidal shape and also confirmed their nanometric size (Figure 1B).

### 2.2. Biocompatibility of Empty and TGF-β1-Loaded Rapeseed Nanoliposomes

The metabolic activity was based on the ability of living cell mitochondria to reduce tetrazolium salt of MTT (3-[4,5-dimethylthiazol-2-yl]-2,5-diphenyltetrazolium bromide) into formazan crystals. Control cells (Ctrl, untreated chondrocytes), cells treated with empty nanoliposomes (NL), and cells treated with nanoliposomes encapsulating TGF-β1 (NL-TGF) with increasing concentrations from 0.1 ng/mL (NL-TGF 0.1) to 100 ng/mL (NL-TGF 100) showed a comparable metabolic activity whatever the growth factor concentration (Figure 2A). Whereas there was no cell proliferation difference between Ctrl, NL and the lowest growth factor concentration NL-TGF 0.1, a significant increase of DNA concentration indicating cell growth was observed for rat chondrocytes cultured with NL-TGF 1, 10, and 100. The highest increase was obtained for NL-TGF 10 with a doubling of the DNA amount compared to Ctrl (Figure 2B). As Ctrl indicates basal lactate dehydrogenase (LDH) release, no significant cytotoxicity between NL and nanoliposomes encapsulating TGF-β1 concentrations from 0.1 ng/mL to 100 ng/mL was detected using the LDH assay (Figure 2C). Taking together, these results demonstrated that the tested nanoliposomes were harmless for rat chondrocytes. Thus, these nanoliposomes are biocompatible for rat chondrocyte culture and do not counteract their viability.

### 2.3. Kinetics of TGF-β1 Release

As shown in the previous section, the condition NL-TGF 10 provided a superior effect in terms of proliferation compared to NL-TGF 100. Consequently, the experimental conditions were reduced and nanoliposomes encapsulating the 3 lowest TGF-β1 concentrations (0.1, 1, and 10 ng/mL) were kept for the next experiments unless indicated otherwise. The amount of TGF-β1 released in cell culture medium by TGF-β1-loaded nanoliposomes was estimated using enzyme-linked immunosorbent assay (ELISA) during a kinetic from 1 h to 96 h. The results showed that whatever the TGF-β1 concentration encapsulated, nanoliposomes can control the release of this growth factor, with a significant release from 3 h which progressively increases until 96 h. The quantity of TGF-β1 released is also proportional to the encapsulated one (Figure 3A–C).

### 2.4. Effect of TGF-β1 Release on Chondrocyte Response

To evaluate the efficacity of TGF-β1 released in cell culture medium by rapeseed nanoliposomes, cells were cultured with different concentrations of TGF-β1 alone (0.1, 1 and 10 ng/mL) or NL-TGF (containing 0.1, 1 and 10 ng/mL). NL-TGF induced the expression of Col 2 (Figure 4A), Sox-9 (Figure 4B), and ACAN (Figure 4C) genes, which are markers of articular phenotype, with a greater effect than nonencapsulated TGF-β1. As the conditions TGF 10 and NL-TGF 1 provided similar results for gene expression (Figure 4A–C), they were used to evaluate the presence of articular ECM components at a global level. Staining with Sirius red and alcian blue allowed us to assess levels of total collagen and glycosaminoglycans (GAG), respectively (Figure 4D,E). NL-TGF 1 significantly increased collagen levels compared to control, contrary to TGF 10 (Figure 4D). NL-TGF 1 and TGF 10 were able to significantly increased GAG levels compared to control with a greater effect for NL-TGF 1 (Figure 4E). These results show that nanoliposomes allow the use of a lower TGF-β1 concentration in rat chondrocyte culture to better maintain their articular phenotype than the growth factor alone.

### 2.5. Effect of TGF-β1 Release on PPi Chondrocyte Production

It has previously been demonstrated that the PPi machinery in chondrocytes (biosynthesis and export) is very reactive to TGF-β1 treatment that induces the expression of genes involved in the production of extracellular PPI (ePPi) such as inorganic pyrophosphate transport regulator (Ank) and ectonucleotide pyrophosphatase/phosphodiesterase 1 (Enpp1) [26]. This statement was used as a gold standard to evaluate the effect of TGF-β1 alone or encapsulated in rapeseed nanoliposomes on Ank and Enpp1 gene expression and on PPi export by rat articular chondrocytes. After a treatment with TGF-β1 alone, a strong expression of Ank and Enpp1 can be observed for TGF 10 while TGF 0.1 had no effect (Figure 5A,B). The gradual liberation of PPi was proportional to the increase in concentration of TGF-β1 alone. (Figure 5C). When TGF-β1 is encapsulated in nanoliposomes, the effect was even more important for NL-TGF 10 but was also present at NL-TGF 0.1 and NL-TGF 1 (Figure 5A,B). The dosage of ePPi confirmed these results (Figure 5C). These results showed that conditions with NL-TGF had a significantly greater effect than those with TGF-β1 alone on the PPi machinery in chondrocytes. Indeed, similar to previous readouts, the conditions TGF 10 and NL-TGF 1 provided a similar effect.

### 2.6. Confirmation of Busy TGF-β1 Signaling Pathways with TGF-β1-Loaded Nanoliposomes

Following their similar effects on chondrocyte phenotype and activity obtained in the two previous sections, the conditions TGF 10 and NL-TGF 1 were chosen to challenge various TGF-β1 signaling pathways. To assess the activation of these pathways, the phosphorylated forms of ERK, p38-MAPK and Smad3/5 were examined. The phosphorylation on p38-MAPK, ERK, and Smad3/5 pathways was significantly increased using NL-TGF 1 or TGF 10 compared to Ctrl or nanoliposomes alone, showing the superior effects of the two conditions with TGF-β1 on these pathways. However, this increase was more efficient using NL-TGF 1 than TGF 10 (Figure 6A). Consequently, the condition NL-TGF 1 was used after TGF-β1 signaling pathways inhibition to evaluate its involvement in chondrocyte behavior. When chondrocytes were stimulated by NL-TGF 1 in the presence of SB203580, a selective p38-MAPK inhibitor, the increase in Ank gene expression was not affected. In contrast, a selective MAPK/ERK kinase 1 (MEK-1) inhibitor that prevents ERK activation, PD98059, reduced the stimulatory effect of TGF-β1 by 60% (Figure 6B, middle panel). Similarly, both ERK and p38 pathway inhibition triggered a nearly 50% decrease of Col 2 gene expression (Figure 6B, left panel). The production of ePPi was not modified by p38 pathway inhibition but fully hindered by the MEK-1 inhibitor (Figure 6B, right panel). Those results demonstrated that the effect of TGF-β1 encapsulated in nanoliposomes is more efficient in vitro than TGF-β1 alone to activate chondrocyte TGF-β1 signaling pathways. If nanoliposomes encapsulating TGF-β1 are used under ERK pathway inhibition, the gene expression of Col2 and Ank are diminished, as well as ePPi production and transport. On the contrary, when the p38 pathway is inhibited, only Col 2 expression is reduced, but there is no influence on Ank or ePPi levels. The results on signaling pathways are consistent with those found in the literature than with the use of TGF-β1 alone [26].

## 3. Discussion

TGF-β is a multifunctional cytokine that controls cell proliferation, cell differentiation and extracellular matrix production. It is of great meaning in development, wound healing, organ fibrosis and tumor metastasis [27]. TGF-β has three subtypes in mammals: TGF-β1, TGF-β2 and TGF-β3 [28]. Activated TGF-β ligands bind to TGF-β receptors on the cell surface and provide the formation of ligand-receptor complexes to initiate signal communication inner the cell. This leads to the activation of Smad proteins and finally transcription regulation of target genes [29].

TGF-β1 is known to stimulate differentiation of mesenchymal stem cells into chondrocytes and to accelerate in vitro synthesis of matrix molecules, such as collagen, proteoglycan and fibronectin [17]. Therefore, TGF-β1 has long been proposed as a potential promoter of cartilage restoration. However, the in vitro use of growth factors in the field of regenerative medicine and particularly for cartilage engineering is limited because of the need for high concentrations to achieve a strong effect, their short stability at 37 °C and the ineffective response due to the burst release for a high dose [13]. To overcome these issues, we proposed to use agro-based nanoliposomes from rapeseed lecithin to protect, transport, and control the release of TGF-β1. The concentration of rapeseed nanoliposomes used to encapsulate TGF-β1 was 100 µg/mL. This was chosen because empty nanoliposomes at this concentration were previously tested for their noncytotoxic effect on rat chondrocytes after evaluation of parameters such as proliferation, LDH release and apoptosis. Rapeseed nanoliposomes were stable when stored at 37 °C for 30 days and showed no influence on the differentiation state of chondrocytes [6]. Moreover, a comparison between empty and TGF-β1-loaded nanoliposomes has already been performed and no significant difference between empty and loaded liposome size, PDI, and ζ-potential has been found [30].

In this article, we synthetized rapeseed liposomes of nanometric size, as measured by DLS and visualized using TEM, with a high negative charge that increases the stability of the formulation, and a small polydispersity. Then, nanoliposomes encapsulating 0.1, 1, 10 or 100 ng/mL of recombinant TGF-β1 were used to treat rat chondrocytes from 24 h to 7 days. None of the conditions were found to be harmful for cell viability revealing these nanoliposomes to be safe for rat chondrocyte in vitro culture. The condition NL-TGF 10 had even a better effect in term of proliferation compared to NL-TGF 100. Moreover, liposomes could potentially decrease the cytotoxicity of active molecules towards chondrocytes as shown by Farmer et al. [31]. In their study, they demonstrated that the bupivacaine loaded in liposomes (EXPAREL^®^) is less chondrotoxic than standard bupivacaine.

To establish the amount of TGF-β1 encapsulation and therefore the dose of TGF-β1 administered to cells, single growth factor ELISA kits were used in the same cell culture conditions (NL-TGF 100 excepted) from 1 h to 96 h. This showed that NL-TGF 0.1, 1 and 10 had a high encapsulation efficiency and that released quantity of growth factor is proportional to the encapsulated one. It also demonstrated the liposomal control and sustained release of TGF-β1 from 3 h to 96 h. The efficacy of TGF-β1 released by nanoliposomes compared to TGF-β1 alone was assessed at the concentrations 0.1, 1 or 10 ng/mL on the in vitro upkeep of chondrocyte articular phenotype and the PPi machinery. Results obtained for TGF-β1 alone were consistent with the literature [26]. They also demonstrated that TGF-β1-loaded nanoliposomes allow the use of a lower TGF-β1 concentration in vitro compared to the growth factor alone and additionally with a better effect to maintain articular phenotype. Indeed, TGF 10 and NL-TGF 1 provided similar results for genes involved in articular phenotype or the production of ePPi, PPi export, GAG synthesis. Additionally, NL-TGF 1 increased collagen levels contrary to NL TGF 10. Those results demonstrated that the encapsulation of TGF-β1 into liposomes allows a sustained and controlled release of TGF-β1 over time, reducing the need for high doses of TGF-β1 alone. Indeed, TGF-β1 alone at a 10 ng/mL concentration is less effective than nanoliposomes encapsulating 1 ng/mL of TGF-β1 (Figure 4, Figure 5 and Figure 6).

Using TGF-β1-loaded nanoliposomes could be a way to reduce the quantity of growth factors needed for experiments, thus influencing the lengths and costs of production and lessening the utilization of resources. Data presented in this study suggest that rapeseed nanoliposomes are highly efficient at delivering low and sustained doses of TGF-β1 required for stimulating cartilage repair, making these delivery systems promising tools for joint regenerative medicine. We showed that low concentrations of NL-TGF gave the same results as higher concentrations of TGF-β1 alone and that these effects were more extended. This proves that rapeseed nanoliposomes were able to successfully protect and deliver TGF-β1, while emphasizing its biofunctionality at low dose.

The specificity of TGF-β1 response was confirmed by using specific TGF-β1 signaling pathway inhibitors showing that the response induced by the controlled release of TGF-β1 encapsulated in nanoliposomes during the exposition of chondrocytes in vitro was similar or significantly higher than the one achieved with TGF-β1 alone. The positive effect of the sustained release of active molecules on chondrocytes was also reported by Ji et al. [32]. In their study, they showed that the sustained release of the anti-inflammatory drug D-glucosamine sulphate from 1,2-distearoyl-sn-glycero-3-phosphocholine liposomes can induce chondrocyte degeneration through the up-regulation of anabolic components and down-regulation of proinflammatory cytokines, pain-related genes, and catabolic proteases, while accelerating the viability and proliferation of cultured chondrocytes.

The in vitro experiments presented here strongly suggest that rapeseed liposomes could be used as natural carriers to deliver other growth factors or bioactive molecules for cartilage repair/regeneration [33,34]. Thus, nanoliposomes could be an attractive alternative to deliver bioactive substances to cartilage. Moreover, the bioactive agro-based NL-TGF developed in this work can have various tissue engineering applications [35,36,37,38,39,40,41].

However, this study has limitations, due to the fact that it was performed on rat chondrocytes and in vitro. Taking these points into account, the results observed herein and the conclusions drawn have to be confirmed for human chondrocytes or in vivo. For these reasons, in future experiments, rapeseed NL-TGF will be tested on human cartilage explant and in vivo directly into the joint to be assured, as observed in vitro in this work, that no cytotoxic effect would be induced and that growth factor biofunctionality would be improved. Since rapeseed liposomes have only recently been developed as a transfection system for chondrocytes [6], they still require extensive testing on small- and large-scale animal models to better understand their interactions within the body, and optimize their safety and efficiency in vivo.

## 4. Materials and Methods

Reagents were obtained from Merck (Saint-Quentin-Fallavier, France) and media components by from Lonza (Colmar, France), unless specified otherwise.

### 4.1. Nanoliposomes Preparation

Rapeseed lecithin was acquired from Solae Europe SA society (Le Grand-Saconnex, Switzerland). A total of 49 mL of distilled water was added to 1 g of rapeseed lecithin and agitated under nitrogen for 5 h. Samples were then probe-sonicated at 40 kHz for 5 min (1 s on, 1 s off) in an ice bath, followed by homogenization using a high-pressure homogenizer (EmulsiFlex-C3, Sodexim SA, Muizon, France). Homogenization was achieved by introducing 50 mL quantities under a pressure of 1500 bar for 7–8 cycles. The produced empty liposomes were stored in glass bottles in the dark at 4 °C until use. To prepare TGF-β1-loaded nanoliposomes, solutions with the appropriate concentrations of recombinant Human TGF-β1 (cat. 100-21, PeproTech, Neuilly-Sur-Seine, France) were made in distilled water and added to rapeseed lecithin then the mixtures were prepared as presented above. The study design illustrating how the produced rapeseed liposomes were used in this publication is summarized in Figure 7.

### 4.2. Physicochemical Characterization

The mean diameter, particle-size distribution and ζ-potential of liposomes were determined upon the dilution of the samples (1:200) using the DLS technique, by employing a Zetasizer Nano ZS (Malvern Instruments Ltd., Worcestershire, UK).

### 4.3. Transmission Electron Microscopy

To monitor the morphology of nanoliposomes, TEM was employed using a negative-staining method as described previously [42]. Briefly, nanoliposomes concentration was reduced by a 30-fold dilution with distilled water. A drop of a solution of 2% ammonium molybdate and the diluted samples was placed for 5 min on a Formvar/carbon supported copper grid (200 mesh, 3 mm diameter HF 36). The mesh was observed using a Philips CM20 operating at 200 kV and micrographs were recorded using an Olympus TEM CCD camera.

### 4.4. Chondrocytes Isolation and Culture

Chondrocytes were isolated from femoral head caps of healthy Wistar male rats (130–150 g) (Charles River, Saint-Aubin-les-Elbeuf, France), euthanized under general anesthesia (AErrane™, Baxter SA, Maurepas, France) according to European animal care guidelines, after approval by our internal ethics committee. Cells were obtained by sequential digestion with pronase and collagenase, then washed twice in phosphate-buffered saline (PBS) and cultured to confluence in 75-cm^2^ flasks at 37 °C in a humidified atmosphere containing 5% CO_2_ as previously described [43]. The medium used was DMEM/Ham’s F-12 supplemented with L-glutamine (2 mM), penicillin (100 IU/mL), streptomycin (100 µg/mL) and either 10% heat-inactivated fetal calf serum (FCS) during subcultures or low FCS medium (1%). Chondrocytes were used at passage 1 and their differentiated articular phenotype was assessed by the high mRNA levels of cartilage specific genes: Col 2, ACAN, and Sox-9. Chondrocytes maintained in low FCS medium were incubated in the presence or absence of empty nanoliposomes (NL), free TGF-β1 (TGF) or TGF-β1-loaded nanoliposomes (NL-TGF), where the control condition is for untreated chondrocytes (Ctrl).

### 4.5. Biocompatibility Assays

To evaluate the impact of empty nanoliposomes and TGF-β1-loaded nanoliposomes on cell behavior, the following parameters were estimated: cell metabolic activity, cell proliferation, and nanoliposome potential cytotoxicity.

#### 4.5.1. Cell Metabolic Activity

Cell metabolic activity was measured using MTT assay as described elsewhere [6]. A total of 50 µL of MTT solution was added to 200 µL of cell culture medium. Briefly, chondrocytes were incubated for 4 h (5% CO2, 95% humidity at 37 °C) to allow the yellow dye to be transformed into insoluble blue formazan crystals by the mitochondrial dehydrogenases. The supernatant was removed and this insoluble product was protected from light and dissolved by addition of 200 µL dimethyl sulfoxide and gently mixed at 37 °C for 5 min. The supernatants were removed, protected from light, centrifuged and their absorbance was read within 30 min using a Varioskan^®^ Flash (Thermo Fisher Scientific, llkirch-Graffenstaden, France) at 540 nm. The control condition for chondrocyte metabolic activity was used as the reference value.

#### 4.5.2. Cell Proliferation

Cell proliferation was assessed using Hoechst assay which allows cell DNA quantification as described elsewhere [6]. Briefly, chondrocytes were harvested from 12-well plates and suspended in 100 µL of Hoechst buffer before 5 series of freezing (liquid nitrogen)/thawing (60 °C, 5 min) cycles for lysing cells and releasing their DNA into solution. Low fluorescent background black fat-bottom plates were used to perform the assay and a calf thymus DNA standard curve was used for the quantification. The samples were mixed with 2 µL of Hoechst solution (0.1 µg/mL in final concentration) and the lecture of DNA sample and standards was performed by fluorescence spectrophotometry (360 nm excitation/460 nm emissions, Varioskan^®^ Flash). The DNA concentration (µg/mL) of each sample was based on its fluorescence measurement relative to the standard curve.

#### 4.5.3. Cytotoxicity Evaluation by LDH Assay

The cytotoxicity test was performed using the Cytotoxicity Detection Kit^PLUS^ (LDH) (Roche, Mannheim, Germany) according to manufacturer’s instructions. This assay is based on the measurement of LDH activity released from the cytosol of damaged cell. Three internal controls are included: background control (assay medium), low control (untreated cells) and high control (maximum LDH release). The absorbance was read on a spectrophotometer at 490 nm (Varioskan^®^ Flash). To determine the experimental absorbance values, the average absorbance values of the triplicate samples and controls were calculated and subtracted from the absorbance values of the background control. The percentage of cytotoxicity was determined over the value of the high control (fixed to 100%).

### 4.6. Evaluation of TGF-β1 Release

Human TGF-β1 was estimated in culture media using Quantikine^TM^ ELISA Kit (cat. DB100B, Bio-Techne-R&D systems, Noyal Châtillon sur Seiche, France). According to manufacturer’s instructions, the sensitivity of the assay in culture was 15.4 pg/mL, the intra-assay precision is < 3%.

### 4.7. RNA Isolation, Reverse Transcription and Real-Time Polymerase Chain Reaction (RT-PCR)

Total RNAs were isolated from cultured chondrocytes, using the Nucleospin RNA kit^®^ (Macherey Nagel, Hoerdt, France) according to the manufacturer’s instructions. Two hundred nanograms of total RNAs were reverse-transcribed for 90 min at 37 °C in a 20 μL reaction mixture containing 10 mM dNTP, 5 μM random hexamer primers, 1.5 mM MgCl_2_, and 200 U Moloney murine leukemia virus reverse transcriptase (Thermo Fisher Scientific, llkirch-Graffenstaden, France). The production of cDNAs was performed in a Mastercycler gradient thermocycler (Eppendorf, Montesson, France). Next, real-time PCR was performed by the Step One Plus^TM^ (Thermo Fisher Scientific, llkirch-Graffenstaden, France) technology using specific primers (Table 1) and iTaq SYBRgreen^TM^ master mix system (Bio-Rad, Marnes-la-Coquette, France). All reagents used for RT-PCR were added at the concentrations recommended by the manufacturer. Melting curve was performed to determine the melting temperature of the specific PCR products and, after amplification; the product size was checked on a 1% agarose gel stained with GelRed^TM^ (Biotium-Interchim, Montluçon, France). Each run included positive and negative reaction controls. The mRNA levels of the gene of interest and of the ribosomal protein 29 (RP29), chosen as housekeeping gene, were determined in parallel for each sample. Relative quantification was determined using the 2^−∆∆Ct^ method and the results were expressed as fold expression normalized over the appropriate control. The calculation formula for fold change or relative fold gene expression level is 2^−∆∆Ct^, with ∆Ct = Ct (gene of interest) − Ct (housekeeping gene) and ∆∆Ct = ∆Ct (treated sample) − ∆Ct (control sample). Ct is the cycle threshold corresponding to the PCR cycle number where the fluorescence generated by the PCR product is perceptible from the background noise. The relative gene expression is set to 1 for control samples because ΔΔCt is equal to 0 and therefore 2^0^ is equal to 1.

### 4.8. Sirius Red and Alcian Blue Staining

Chondrocytes were plated in 12-well dishes and cultured for 7 days. Cells were rinsed with PBS, fixed for 30 min in 4% paraformaldehyde and stained with a Sirius red solution (0.1% Sirius red dissolved in saturated picric acid) for 1 h or Alcian blue solution (0.1% Alcian blue in 0.1 M HCl, pH 1) overnight.

Quantification of Sirius Red was obtained at 550 nm (Varioskan^®^ Flash) after coloration dissolution in 0.1 M NaOH. To quantify Alcian blue, staining was dissolved in 4 M HCl and absorbance was read at 600 nm (Varioskan^®^ Flash). Results were normalized over the control condition.

### 4.9. Radiometric Assay for Extracellular Inorganic Pyrophosphate (ePPi)

Extracellular PPi levels were measured using the differential adsorption of uridine diphospho-(6-^3^H) glucose (GE Healthcare, Buc, France), and its reaction product 6-phospho-(6-^3^H) gluconate on activated charcoal, as previously described [44]. The standard concentrations, ranging from 10 to 400 pmol of ePPi, were included in each assay. After adsorption of the reaction mixture on charcoal, and centrifugation at 14,000 rcf for 10 min, 100 μL of supernatant were removed carefully and counted for radioactivity in 5 mL of Bio-Safe II (Research Products International Corp., Mt. Prospect, IL, USA). Results were expressed as picomoles of ePPi per microgram of total cell proteins.

### 4.10. Protein Extraction and TGF-β1 Signaling Pathways Analysis

Proteins were analyzed by Western blot. Total proteins were extracted using Laemmli buffer, separated on 4-20% sodium dodecyl sulphate-polyacrylamide gel electrophoresis and then transferred to nitrocellulose membrane. After 1 h in blocking buffer (tris-buffered saline (TBS)-Tween^®^ 20-5% skim milk), membranes were washed 3 times with TBS-Tween^®^ 20 and incubated overnight at 4 °C with primary antibodies for p-ERK, p-p38-MAPK, p-Smad3/5, β-actin (Cell signaling Technology-Ozyme, Saint-Cyr-L’École, France) used at 1:1000 dilution in blocking buffer. After three washing steps with TBS-Tween^®^ 20, blots were incubated for 1 h at room temperature with antirabbit IgG conjugated with horseradish peroxidase (Cell Signaling Technology-Ozyme, Saint-Cyr-L’École, France) at 1:4000 dilution in blocking buffer. After three washing steps with TBS-Tween^®^ 20, signal was detected by chemiluminescence (Clarity^TM^ western ECL substrate, Bio-Rad, Marnes-la-Coquette, France).

When used, specific pharmacological inhibitors were added 1 h before stimulation with NL-TGF at the following concentrations: 10 µM for SB203580 (a selective p38- MAPK inhibitor) and 30 µM for PD98059 (a MEK-1 inhibitor that prevents ERK activation).

### 4.11. Statistical Analysis

Results are expressed as the mean ± SD. Statistical analysis were performed with GraphPad Prism 6 (GraphPad Software, San Diego, CA, USA) using one-way ANOVA multiple comparisons followed by Tukey correction. *p* values were indicated in the legends if considered significant with *p* < 0.01.

## Figures and Tables

**Figure 1 ijms-23-02864-f001:**
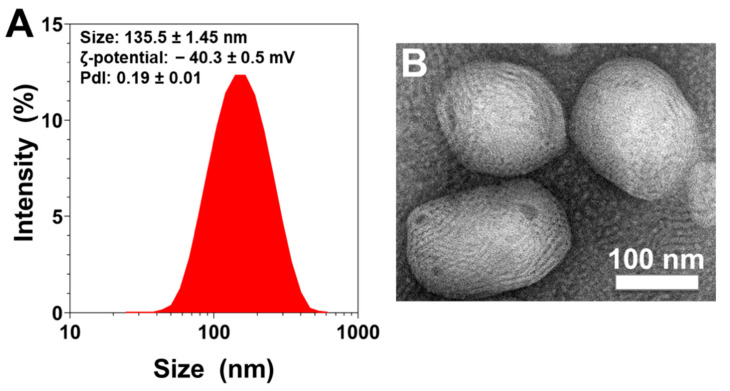
Physical evaluation of rapeseed liposomes. (**A**) Size distribution measurements, and average size, ζ-potential, and polydispersity index values of liposomes measured by dynamic light scattering. The reported data are represented as mean ± SD of at least three individual experiments. (**B**) Representative transmission electron microscopic picture of liposomes. The white scale bar corresponds to 100 nm. PDI—polydispersity index; SD—standard deviation.

**Figure 2 ijms-23-02864-f002:**
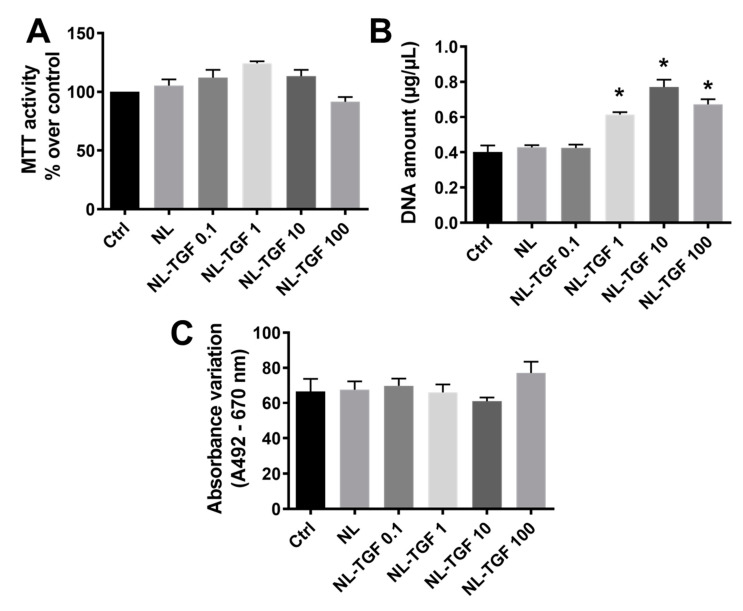
Impact of rapeseed nanoliposomes encapsulating or not encapsulating TGF-β1 on chondrocyte viability. Cells were untreated or exposed to 100 µg/mL of nanoliposomes encapsulating or not 0.1, 1, 10 or 100 ng/mL of recombinant TGF-β1 for 24 h or 7 days. (**A**) Metabolic activity measured using MTT assay at day 7. The control condition was used as the reference value corresponding to an activity of 100%. (**B**) Cell proliferation estimated by measuring DNA concentrations at day 7. (**C**) In vitro cytotoxicity measured by LDH assay at day 1. The reported data are represented as mean ± SD of at least four individual experiments. Significance compared to Ctrl is indicated as * with *p* < 0.01. Ctrl—control cells meaning untreated chondrocytes; LDH—lactate dehydrogenase; MTT—3-(4,5-dimethylthiazol-2-yl)-2,5-diphenyltetrazolium bromide; NL—empty nanoliposomes; NL-TGF—TGF-β1-loaded nanoliposomes; NL-TGF 0.1—nanoliposomes encapsulating 0.1 ng/mL of TGF-β1; SD—standard deviation; TGF-β1—transforming growth factor-β1.

**Figure 3 ijms-23-02864-f003:**
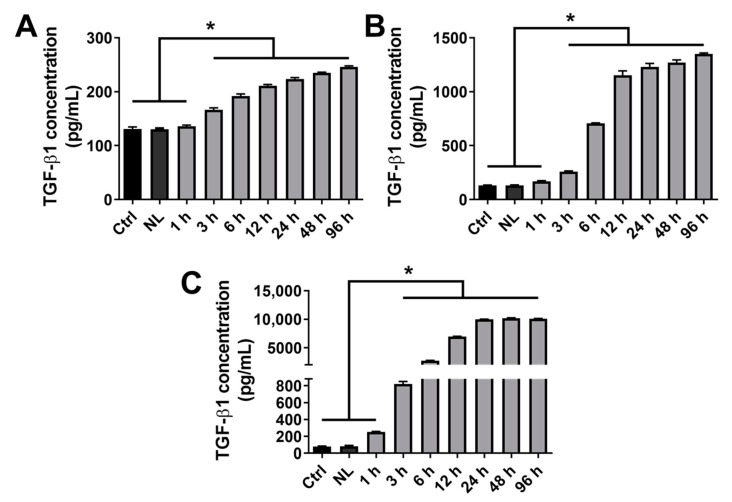
Kinetics of TGF-β1 release from rapeseed nanoliposomes in cell culture medium. Cells were untreated or exposed to 100 µg/mL nanoliposomes, either encapsulating or not 0.1 (**A**), 1 (**B**) or 10 (**C**) ng/mL of recombinant TGF-β1 from 1 h to 96 h. The release of TGF-β1 in cell culture media was quantified by ELISA. The reported data are represented as mean ± SD of at least four individual experiments. Significance is indicated as * with *p* < 0.01. Ctrl—control cells meaning untreated chondrocytes; ELISA—enzyme-linked immunosorbent assay; NL—empty nanoliposomes; NL-TGF—TGF-β1-loaded nanoliposomes; NL-TGF 0.1—nanoliposomes encapsulating 0.1 ng/mL of TGF-β1; SD—standard deviation; TGF-β1—transforming growth factor-β1.

**Figure 4 ijms-23-02864-f004:**
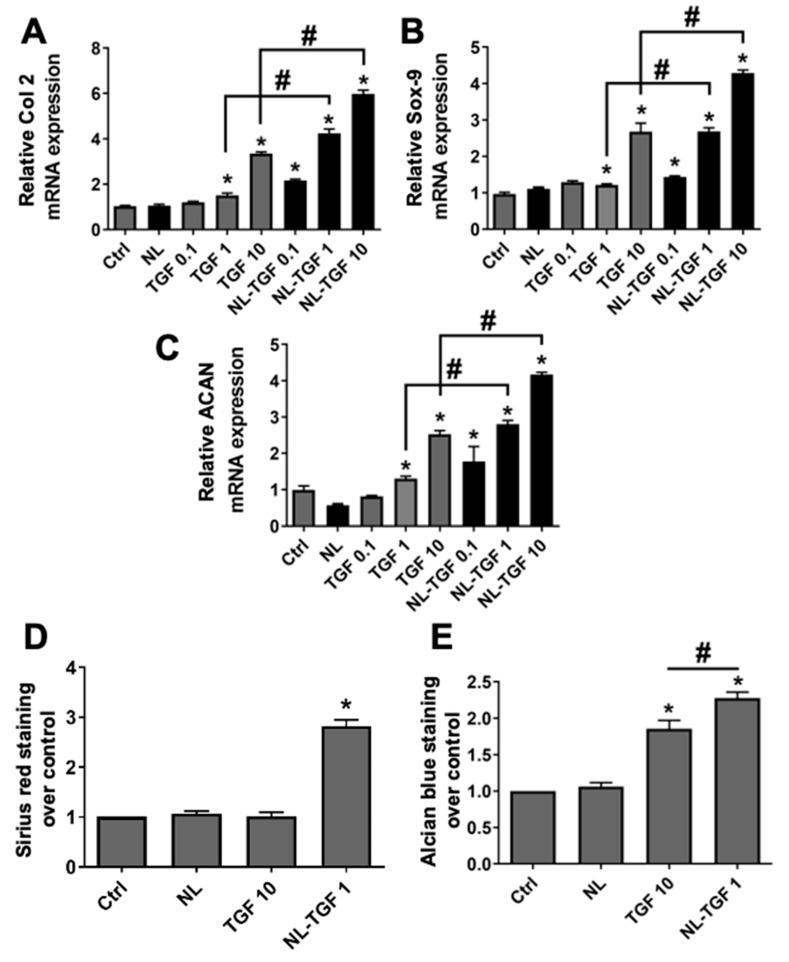
Efficacity of TGF-β1 release on chondrocyte articular phenotype. Rat chondrocytes were untreated or exposed to 100 µg/mL of nanoliposomes encapsulating or not encapsulating TGF-β1 (0.1, 1 or 10 ng/mL), or to TGF-β1 alone (0.1, 1 or 10 ng/mL) for 24 h (mRNA) or 7 days (Sirius red and alcian blue staining). Total RNA was extracted then reverse transcribed into cDNA and analyzed by real-time PCR. The relative abundance of (**A**) Col 2, (**B**) Sox-9, or (**C**) ACAN was normalized to RP29. Quantifications were made by using the ΔΔCt method. (**D**) Sirius red and (**E**) alcian blue staining were made to quantify respectively collagens and glycosaminoglycans. Results were normalized over Ctrl. The reported data are represented as mean ± SD of at least four individual experiments. Significance is indicated as * when compared to Ctrl and # (*p* < 0.01). ACAN—aggrecan; Col 2—type II collagen; Ctrl—control cells meaning untreated chondrocytes; NL—empty nanoliposomes; NL-TGF—TGF-β1-loaded nanoliposomes; NL-TGF 0.1—nanoliposomes encapsulating 0.1 ng/mL of TGF-β1; PCR—polymerase chain reaction; RP29—ribosomal protein 29; SD—standard deviation; Sox-9—SRY (Sex-Determining Region Y)-Box 9; TGF-β1—transforming growth factor-β1; TGF 0.1—TGF-β1 alone at a concentration of 0.1 ng/mL.

**Figure 5 ijms-23-02864-f005:**
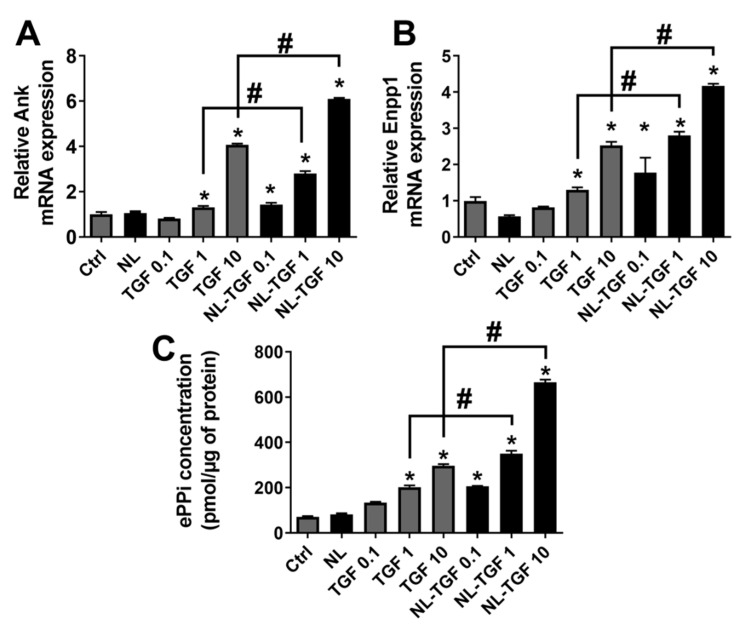
Efficacity of TGF-β1 release on PPi chondrocyte production. Rat chondrocytes were untreated or exposed to 100 µg/mL of nanoliposomes encapsulating or not encapsulating TGF-β1 (0.1, 1 or 10 ng/mL), or to TGF-β1 alone (0.1, 1 or 10 ng/mL) for 24 h (mRNA) or 48 h (PPi). Total RNA was extracted then reverse-transcribed into cDNA and analyzed by real-time PCR. The relative abundance of (**A**) Ank and (**B**) Enpp1 was normalized to RP29. Quantifications were made by using the ΔΔCt method. (**C**) ePPi levels in culture supernatant of rat chondrocytes were assayed radiometrically and normalized to the amount of total cell proteins. The reported data are represented as mean ± SD of at least four individual experiments. Significance is indicated as * when compared to Ctrl and # (*p* < 0.01). Ank—inorganic pyrophosphate transport regulator; Ctrl—control cells, meaning untreated chondrocytes; Enpp1—ectonucleotide pyrophosphatase/phosphodiesterase 1; ePPi—extracellular inorganic pyrophosphate; NL—empty nanoliposomes; NL-TGF—TGF-β1-loaded nanoliposomes; NL-TGF 0.1—nanoliposomes encapsulating 0.1 ng/mL of TGF-β1; PCR—polymerase chain reaction; PPi—inorganic pyrophosphate; RP29—ribosomal protein 29; SD—standard deviation; TGF-β1—transforming growth factor-β1; TGF 0.1—TGF-β1 alone at a concentration of 0.1 ng/mL.

**Figure 6 ijms-23-02864-f006:**
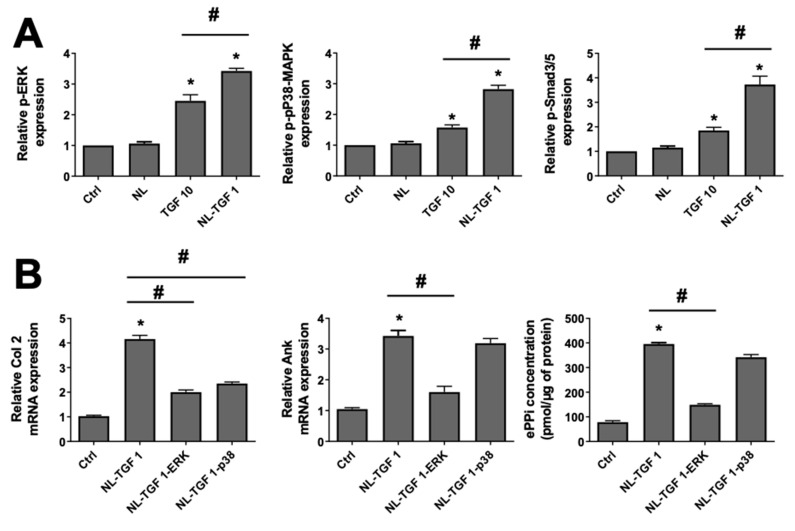
Activation status of TGF-β1 signaling pathways on chondrocyte. (**A**) Signaling events induced by TGF-β1. Rat chondrocytes were untreated or exposed to 100 µg/mL of nanoliposomes encapsulating or not encapsulating TGF-β1 (1 ng/mL), or to TGF-β1 alone (10 ng/mL) for 10 min (p-ERK and p-p38-MAPK) or 3 h (p-Smad3/5). Total proteins were extracted from cells and subjected to Western blotting using anti-phospho-ERK, anti-phospho-p38-MAPK or anti-phospho-Smad3/5. The relative abundance of these proteins was normalized to that of β-actin protein. The reported data are represented as mean ± SD of at least three individual experiments. Significance is indicated as * when compared to Ctrl and # when compared to TGF-β1 alone (*p* < 0.01). (**B**) Rat chondrocytes were exposed to 100 µg/mL of nanoliposomes encapsulating or not encapsulating TGF-β1 (1 ng/mL) for 24 h (mRNA) or 48 h (PPi), and with or without 10 μM of SB203580 (a selective p38-MAPK inhibitor) or 30 μM of PD98059 (a MEK-1 inhibitor) added 1 h before TGF-β1. Total RNA was extracted then reverse transcribed into cDNA and analyzed by real-time PCR. The relative abundance of Col 2 and Ank was normalized to RP29. Quantifications were made by using the ΔΔCt method. ePPi levels in culture supernatant of rat chondrocytes were assayed radiometrically and normalized to the amount of total cell proteins. The reported data are represented as mean ± SD of at least three (mRNA) or six (ePPi) individual experiments. Significance is indicated as * when compared to NL and # (*p* < 0.01). Ank—inorganic pyrophosphate transport regulator; Ctrl—control cells, meaning untreated chondrocytes; ePPi—extracellular inorganic pyrophosphate; ERK, extracellular signal-regulated kinase; MEK-1, mitogen-activated protein kinase/extracellular signal-regulated kinase kinase 1; NL—empty nanoliposomes; NL-TGF—TGF-β1-loaded nanoliposomes; NL-TGF 1—nanoliposomes encapsulating 1 ng/mL of TGF-β1; NL-TGF 1-ERK—condition NL-TGF 1 after ERK pathway inhibition (PD98059); NL-TGF 1-p38—condition NL-TGF 1 after p38 pathway inhibition (SB203580); p38-MAPK—p38 mitogen-activated protein kinase; PCR—polymerase chain reaction; p-ERK—phosphorylated ERK or phospho-ERK; p-P38—phosphorylated p38 or phospho-P38; p-Smad3/5—phosphorylated pSmad3/5 or phospho- pSmad3/5; RP29—ribosomal protein 29; SD—standard deviation; TGF-β1—transforming growth factor-β1; TGF 10—TGF-β1 alone at a concentration of 10 ng/mL.

**Figure 7 ijms-23-02864-f007:**
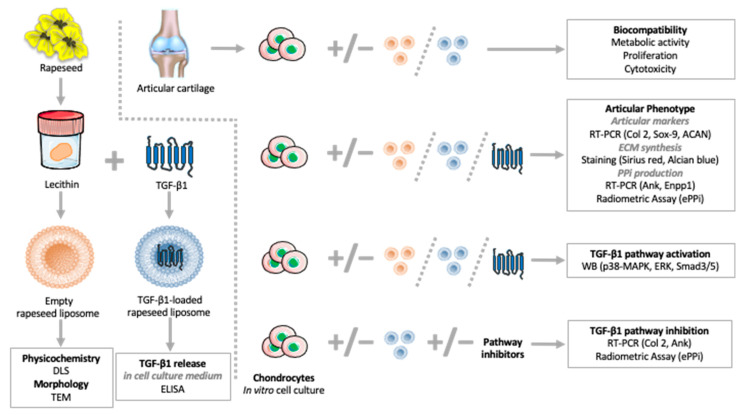
Study design summary of rapeseed liposome production and experimental use. ACAN—aggrecan; Ank—inorganic pyrophosphate transport regulator; Col 2—type II collagen; DLS—dynamic light scattering; ECM—extracellular matrix; ELISA—enzyme-linked immunosorbent assay; Enpp1—ectonucleotide pyrophosphatase/phosphodiesterase 1; ePPi—extracellular inorganic pyrophosphate; ERK, extracellular signal-regulated kinase; p38-MAPK—p38 mitogen-activated protein kinase; PPi—inorganic pyrophosphate; RT-PCR—real-time polymerase chain reaction; Sox-9—SRY [Sex-Determining Region Y]-Box 9; TEM—transmission electron microscopy; TGF-β1—transforming growth factor-β1.

**Table 1 ijms-23-02864-t001:** Sequences of specific primers for RT-PCR analyses.

Genes	Sequences 5′-3′
ACAN	Fwd: CAA-CCT-CCT-GGG-TGT-AAG-GA
	Rev: TGT-AGC-AGA-TGG-CGT-CGT-AG
Ank	Fwd: CAA-GAG-AGA-CAG-GGC-CAA-AG
	Rev: AAG-GCA-GCG-AGA-TAC-AGG-AA
Sox-9	Fwd: CTG-AAG-AAG-GAG-AGC-GAG-GA
	Rev: GGT-CCA-GTC-ATA-GCC-CTT-CA
Col 2	Fwd: TCC-CTC-TGG-TTC-TGA-TGG-TC
	Rev: CTC-TGT-CTC-CAG-ATG-CAC-CA
Enpp1	Fwd: TAT-GCC-CAA-GAA-AGG-AAT-GG
	Rev: GCA-GCT-GGT-AAG-CAC-AAT-GA
RP29	Fwd: CTC-TAA-CCG-CCA-CGG-TCT-GA
	Rev: ACT-AGC-ATG-ATT-GGT-ATC-AC

ACAN—aggrecan; Ank—inorganic pyrophosphate transport regulator; Sox-9—SRY (Sex-Determining Region Y)-Box 9; Col 2—type II collagen; Enpp1—ectonucleotide pyrophosphatase/phosphodiesterase 1; Fwd—forward primer; RT-PCR—real-time polymerase chain reaction; Rev—reverse primer; RP—ribosomal protein.

## Data Availability

Not applicable.

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
