# Peer review of "Efficient TGF-β1 Delivery to Articular Chondrocytes In Vitro Using Agro-Based Liposomes"

_ijms, 2022, doi:10.3390/ijms23052864_

Round 1

Reviewer 1 Report

The authors did a thorough study in the evaluation of TGF-β1 delivery with agro-based liposomes. This article discussed the biocompatibility of nanoliposome carrier, kinetic of cargo release and expression levels of biomarkers representing chondrocyte response. Potential pathways TGF-β1-loaded nanoliposomes utilized to function were also involved in this article. Overall, the manuscript is comprehensive, well-written and likely to appeal to a broad readership with an interest in the field of application of nanoliposome for delivery of small molecules in joint regenerative therapeutics. I suggest minor revision before publication. Here attached questions and comments to the authors.

  1. This manuscript is generally well-organized. A little bit more description of indication/significance of this study may be helpful. Also, the readers may benefit from additional rationale of biomarkers/kinases chose to test in this study.
  2. I’m curious how this nanoliposome perform in in vivo studies. Liposomes can usually interact with plasma proteins, which leads to opsonization, therefore effecting the healthy cells they come into contact with during circulation and removal. How stable will the liposome in the circulation system? Due to the pharmacokinetics of liposomes in circulation, drugs can end up accumulated in organs of the mononuclear phagocyte system, affecting liver and spleen function. Do the authors have any preliminary data or literature search to evaluate the potential toxicity, stability and efficiency of the proposed delivery system?
  3. The authors mentioned that they used primary chondrocytes isolated from rats. How long can these chondrocytes be cultured at most? How the epitopes change along with the culture? I would suggest the authors to provides data at a longer time point for the cytotoxicity of a nanoparticle.
  4. Although the results of cell proliferation study indicated that cell growth was observer, is 24 hr enough for nanoliposomes to execute cytotoxicity if there’s any regarding the cytotoxicity study?
  5. Could the authors provides more illustration of the ∆∆Ct method for real-time PCR quantification?

Reviewer 2 Report

Velor and co-workers reported the encapsulation of TGF-β 1 into rapeseed liposomes for the delivery to articular chondrocytes. The work carried outcomes from the continuation of an article previously published by the authors (reference 6) in which the production of liposomes and their effect on chondrocytes was performed. In this study, the authors encapsulated TGF-β1 and evaluated its effect. Several data is missing and needs to be introduced before acceptance of the paper.

First, I believe that the study regarding size, zeta-potential, PDI, and morphology (section 2.1), should be carried with the TGF-β1 encapsulated to observe any differences compared to the empty liposomes. The results presented (only the empty liposomes) are not important in this paper, regarding that are already published. Moreover, since TGF-β1 is a protein and not a small molecule, is crucial to present its impact on the structure of the formulated liposomes.

The authors do not use any separation device or procedure to eliminate unencapsulated TGF-β1. Since it’s a large macromolecule, the results of activity obtained may be influenced by the “free” TGF-β1 and not derivate from the encapsulated one. Is not accurate to test the activity of TGF-β1 if a previous separation process is not performed.

Some information regarding the stability (of storage) of the encapsulated liposomes should also be provided.

Minor:

Figure 6 is not focused. Please replace it.

Reviewer 3 Report

In this article by Émilie Velot et colleagues, the authors evaluated naturally-derived liposomes as promising delivery nanosystems due to their similarities with biological membranes. The authors used agro-based rapeseed liposomes, which contains a high level of mono- and poly-unsaturated fatty acids, to efficiently deliver encapsulated TGF-β1 to rat chondrocytes.

The MS is well organized, I have the next suggestions for improving it:

A summary diagram with all the steps of this complex study is recommended.

Add in the Discussion section, the following aspects:  comparisons with similar studies in the literature, limitations. What potential clinical gaps for humans does this MS have?

Consider revision accordingly.

Round 2

Reviewer 2 Report

The authors clearly responded to my comments and suggestion. I suggest that the information responded should be included in the manuscript with the respective references.
